# The Value of Ultrasound Diagnostic Imaging of Anterior Crucial Ligament Tears Verified Using Experimental and Arthroscopic Investigations

**DOI:** 10.3390/diagnostics14030305

**Published:** 2024-01-30

**Authors:** Cezary Wasilczyk

**Affiliations:** Medical Department, Wasilczyk Medical Clinic, ul. Kosiarzy 37/80, 02-953 Warszawa, Poland; wasilczyk.chirurg@gmail.com

**Keywords:** anterior crucial ligament, anterior crucial ligament tears, ultrasound, diagnostic imaging, arthroscopy

## Abstract

This study investigates the potential of the ultrasound imaging technique in the assessment of Anterior Cruciate Ligament (ACL) pathologies by standardizing the examination process. We focused on four key ultrasound parameters: the inclination of the ACL; swelling or scarring at the ACL’s proximal attachment to the lateral femoral condyle; swelling or scarring of the ACL/posterior cruciate ligament (PCL) compartment complex with accompanying morphological changes in the posterior joint capsule; and dynamic instability, categorized into three ranges—0–2 mm, 3–4 mm, and ≥5 mm. The study group consisted of 25 patients with an ACL injury and 25 controls. All four tested parameters were found more frequently in the study group compared to the control (*p* < 0.0001). Our findings suggest that this standardized approach significantly augments the diagnostic capabilities of ultrasound, complementing clinical evaluation and magnetic resonance imaging (MRI) findings. The meticulous assessment of these parameters proved crucial in identifying subtle ACL pathologies, which might otherwise be missed in conventional imaging modalities. Notably, the quantification of dynamic instability and the evaluation of morphological changes were instrumental in early detection of ACL injuries, thereby facilitating more precise and effective treatment planning. This study underscores the importance of a standardized ultrasound protocol in the accurate diagnosis and management of ACL injuries, proposing a more comprehensive diagnostic tool for clinicians in the field of sports medicine and orthopedics.

## 1. Introduction 

The anterior cruciate ligament (ACL) is crucial to restrain anteroposterior rotational and derivative stabilization of the human knee. Knee instability due to an ACL tear is the first step in a cascade of events leading to rotational and anterolateral instability, meniscal and chondral lesions, and, ultimately, to osteoarthritis [1,2]. An increasing interest in sports activities causes a higher percentage of knee injuries, including ACL tears.

Over 70% of ACL injuries occur during sports activities. The remaining 30% of cases occur as a result of road traffic accidents and accidents at work. Among contact sports, soccer and volleyball carry a high risk of ACL injury. Among non-contact sports, the highest risk is reported for skiing and acrobatics. ACL tears due to energy impact may also be divided into low-energy injury (during sports activities) and high-energy injury (falls from a high level or road traffic accidents) [3,4].

ACL tears account for 40–50% of all ligamentous knee injuries. In sports activities, women display a 6-fold higher frequency of such injuries than men [5,6]. Moreover, ACL injuries are more common in the young athletic population than in adults. 

Dodson et al. documented 219 ACL injuries in National Football League players between 2010 and 2013. Forty players (18.3%) had previous ACL injuries, with 27 retears (12.3%) and 16 players (7.3%) suffered contralateral limb ACL injury. The authors concluded that retears and contralateral ACL tears constituted a substantial percentage of all ACL injuries in NFL players [7].

During a follow-up after five years, John et al. reported that in a group of 465 Indian athletes with a history of knee injuries, there were 314 patients with ACL tears. Soccer was the most common sport activity associated with knee injury [3], and the most common mechanism was non-contact injuries. The mean duration of time lost in sport activities among those who returned to sport was 8.84 months, and only 39.8% of the patient group returned to sport [8].

New technologies make it possible to assess the ACL anatomic footprint areas, biological structure, function, and mechanical resistance to injury [9,10].

The ACL courses anteriorly and medially across the knee joint, reaching the distal end at the tibial surface. The role of the ACL is to restrain anteromedial and rotational translation between the femur and tibia. Secondarily, the ACL is involved in the varus and valgus stability of the knee joint. The ACL consists of two bundles: anteromedial and posterolateral, named for their tibial insertion area [11,12]. The anatomical ACL division of the two bundles is also reasonable in the functional aspect. The anteromedial bundle stabilizes anteromedial translation, and the posterolateral bundle stabilizes rotational translation in the knee joint [13,14,15,16]. Ferratti et al. described two bundles in the ACL structure in early fetal development. Microscopically, the ACL consists of collagen type III arranged longitudinally to the long axis of the ligament. The blood supply originates from the middle genicular artery from the popliteal artery. Innervation of the ACL comes from the posterior articular nerve: the branch of the tibial nerve [17,18]. 

High-risk maneuvers causing ACL tears include forced valgus, varus, hyperextension, or hyperflexion, or any combination thereof. Rarely, the mechanism of an ACL tear is high-energy direct injury. The risk factors for an ACL tear include sex, body mass index, age, previous knee injuries, current knee complaints, family disposition towards an ACL injury, and neuromuscular or neurological dysfunction. Female athletes sustained ACL injuries at a higher rate than male athletes. Family history, patients with knee complaints, and a high body mass index also play a part in increased risk of an ACL tear [19].

Ristić et al. assessed risk factors for bilateral ACL injuries. In a group of 32 patients undergoing surgery for a bilateral ACL injury, the frequency of injury did not depend on sex, side of extremity, genetic predisposition, type of sport, concomitant injuries, and the choice of graft. This collateral type of injury most often occurs in young athletes in the first four years following contralateral reconstruction. The incidence of reconstructed bilateral vs unilateral ACL tears in the study was 2.3%—50 athletes in the group of 2168 patients [20]. In a study by Dodson et al., an important external risk factor for ACL injury was the position of the football player on the sports field. Other important external risk factors are the environmental conditions, such as the type of background or shoes. Important ACL tear factors are also the personal biologic and mechanic features of the knee and soft and bone tissue structure [21,22,23].

Roosi et al. described the tibial tubercle–trochlear groove distance as an independent risk factor in non-contact ACL injury. According to the author, a 2 mm increase in this distance seems to be associated with a higher risk of an ACL tear [24]. Degnan et al. confirmed a significant association of an ACL tear with the Insall–Salvati ratio in pediatric patients [25]. Moreover, Weiss et al., in a literature review of 15 articles, found a relationship between patellofemoral pain and an ACL tear [26]. In a systematic review, John et al. assessed whether there was evidence for a genetic predisposition to ACL tears. In 14 of the 17 studies, the authors focused on specific gene polymorphisms, of which different polymorphisms of 10 genes were positively associated with an ACL tear [27].

### 1.1. ACL Imaging Mmethods

To confirm an ACL tear and to exclude or confirm other knee joint injuries, such as fracture, meniscal tears, or lose bodies inside the joint capsule, imaging of the post-injury knee has to be performed. The imaging tests include ultrasound, X-ray, and magnetic resonance imaging. Plain radiographic imaging helps exclude or confirm fractures, including avulsion ones [28,29]. The standardized MRI protocol consists of sagittal, coronal, and axial planes that help determine the ACL structure. Numerous studies proved the high sensitivity and specificity of the MRI based on arthroscopy findings [30,31,32,33,34,35]. Using MRI views, the Blumensaat angle is helpful to diagnose ACL injuries [36]. Ultrasound examination plays an important role in post-traumatic knee evaluation. Wang et al. assessed four studies involving 246 patients. The combined sensitivity and specificity were 90.0% (95% CI: 77–96) and 97%, respectively [37]. The evaluation of the knee joint swelling including extra- and intra-articular compartments, differential diagnosis of ligamentous and meniscal injury, and dynamic examination, which is a very strong tool. Extra- and intra-articular injuries shown on an ultrasound examination are well documented with a high rate of positive diagnoses [38,39,40,41,42,43]. 

### 1.2. ACL Clinical Evaluation

The clinical evaluation of the ACL consists of reviewing medical history, physical examination, and clinical tests. Medical history should provide detailed information about the mechanism and circumstances of the injury. Information such as time from injury, ability to continue activities, joint swallow, blocking, or giving away help when making a differential diagnosis [44,45,46]. Clinical examination and testing involve the Lachman test, pivot shift test, and anterior drawer test. Clinical testing with instability grading should be performed by an experienced surgeon. In an acute ACL examination, it is important to conduct the evaluation with the minimal possibility of an increase in pain. The Lachman test is supposed to be performed in 20-degree flexion with the end point evaluation [47,48,49]. Anterior laxity with no end point or soft end point in the Lachman test is often a positive test for an anteromedial bundle ACL tear. The pivot shift test and its derivatives test are important clinical tests to determine ACL function. With the knee in full extension, valgus stress is placed on the tibia, while an axial load, internal rotation, and flexion are applied. If, during the motion, the lateral tibia plateau subluxates to the lateral femur condyle and during further flexion the lateral tibia is reduced, the pivot shift test is positive. The rotational instability in most cases is more effective for the patient than anterior instability [50,51,52]. There are many factors that may influence the clinical evaluation of the ACL, including pain restriction, edema scarification inside the joint including ACL remnants scarification, hip position, and tibia rotation. The pivot shift test demonstrated the highest degree when the hip is abducted, and the tibia is externally rotated. The anterior drawer test in clinical finding is the next specific test in the clinical evaluation of the ACL [53]. Colombet et al. reported a significant difference between a total and a partial ACL tear based on clinical examination [54]. Some authors proved that the sensitivity of a clinical test is increased in the case of chronic ACL tears [55]. If hemarthrosis is present beside an ACL injury, patellofemoral joint injury, meniscus tear, or chondral lesion should be considered [56,57,58].

### 1.3. ACL Injury Types and Treatment

ACL injury division may depend on clinical and imaging findings or on arthroscopic assessment. The division of structural ACL injuries may depend on (1) total or partial tear (with subdivision of anteromedial and posterolateral or of both tears); (2) proximal, distal, or intrafiber rapture; and (3) isolated or concomitant ACL injury—multiple ligamentous, meniscal, and osteochondral injuries [59,60,61]. Noyes defines total ACL tears as those in which over 75% of the ligament fibers are torn [62]. Moreover, Hong et al. defined partial ACL depending on the percentage of ACL fiber tears. According to the authors, partial ACL injury is defined as being when less than 50% is torn [63,64,65]. The diagnosis of partial ACL tears is still controversial and challenging for orthopedic surgeons. The incidence of partial tears ranges from 9% to over 27% [59]. So far, the difference between a partial or total ACL tear is unclear. Some authors define an ACL tear based on operational findings, while others base it on clinical or imaging evaluations. The American Medical Association divides ACL injury into three degrees: the first and second degrees are defined as a partial tear and the third degree is defined as a complete tear of the ACL [59].

### 1.4. Aims

The main goal of this study is to objectify ultrasound imaging diagnostics of ACL knee injuries based on individual features of ultrasound presentation and to evaluate the applicability of this modality in clinical practice.

The specific objectives of the study were:To identify key parameters for ultrasound examination of ACL tears.To assess the accuracy of ultrasound in the evaluating of ACL tears.To determine ultrasound examination accuracy of anatomical parts, including arthroscopy procedure.To provide a statistical assessment of the characteristic features of ultrasound ACL tears for clinical application.An attempt at standardization of ultrasound examination of the ACL.

## 2. Materials and Methods

The study was divided into an anatomical and a clinical part. The study was approved by the Bioethical Committee in Warsaw, Poland (No. 52/21). All procedures were performed in compliance with the Declaration of Helsinki. 

An attempt to standardize the ultrasound examination of the ACL was made by evaluating four described parameters. 

The inclination of the ACL—patient is in supine position with a knee flexion of 90 degrees. The transducer is applied along the sagittal plane parallel to the longitudinal axis of patellar ligament. Inclination angle is the angle between base line and front line of the ACL.Swelling/scarifications of the ACL proximal attached to lateral femoral condyle—the patient is in prone position with knee full extension. The transducer is applied transversely to long axis of the lower limb in popliteal fossa.Swelling/scarifications of the ACL/posterior crucial ligament (PCL) with change of the morphology of the posterior joint capsule complex—the patient is in prone position with knee full extension. The transducer is applied parallel to the long axis of the lower limb.Dynamic instability—patient is in supine position with a knee flexion of 90 degrees. The transducer is applied along the sagittal plane parallel to the longitudinal axis of patellar ligament with dynamic anterior drawer test.

This standardization was based on the data from 20 patients, each of whom had undergone an ultrasound examination and had a clinically and MRI-confirmed ACL injury. The patient group comprised 8 men and 12 women, aged between 20 and 45 years.

### 2.1. Part I—Anatomical Study

This part of the study was carried out using 10 specimens of a normal ACL obtained from 10 deceased patients (6 men and 4 women). The material for the anatomical part was provided by the collection of the Department of Descriptive and Clinical Anatomy at the Medical University of Warsaw.

Ten knee specimens without an ACL injury on ultrasound were qualified for arthroscopy procedure. The normal presentation of the ACL was confirmed in arthroscopy view of the examined specimen. Next, the ACL injury was performed according to known ACL injury patterns. In the final stage, the injured ACL was reassessed using ultrasound imaging. The study was carried out by two individuals. The ultrasound technician who performed the evaluation of ACL injuries had no knowledge of the mechanism of a particular injury or the actual morphology of an ACL tear. 

Ultrasound evaluation was standardized and performed using a 12 MHz linear transducer. Ultrasound images were stored.

### 2.2. Part II—Clinical Study

This part of the study included 50 participants. The sample size was determined based on the expectation that abnormal ultrasound parameters would be present in 25% of patients in the control group and in at least 75% of the study group. This calculation assumes a significance threshold of alpha = 0.05. Based on these parameters, the ideal sample size for statistical significance in both the study and control groups should have been 19 patients each. However, considering the relative ease of patient recruitment, it was decided to include 25 patients in each group to strengthen the study’s robustness and potentially increase the reliability of the results.

The study group consisted of 25 patients (13 men and 12 women, aged 21–44 years) with ACL injuries, and 25 control subjects (11 men and 14 women, aged 19–47 years). The inclusion criteria were as follows: (1) traumatic history of knee injury; (2) a positive clinical test for anterior or rotational instability; and (3) confirmation of ACL injury through MRI. The exclusion criteria for the study were defined as follows: accompanying intra-articular knee joint injuries that necessitated emergency surgery, such as an unstable meniscus injury of the knee joint with joint blockage, chondral injury involving the separation of a cartilaginous fragment larger than 1 cm, or the presence of loose bodies in the joint exceeding 1 cm in size. Additionally, pregnancy and breastfeeding were also criteria for exclusion from the study.

Controls were patients who underwent ultrasound examination and MRI due to injuries unrelated to ACL tears.

First, clinical examination was performed, including the pivot shift test and Lachman test. Subsequently, patients with positive results on clinical examination underwent ultrasound examination. Finally, MRI was performed on all patients. Clinical and ultrasound evaluation was performed by two physicians: an experienced clinician and an experienced ultrasound technician. The ultrasound technician did not take part in clinical examination. Ultrasound examinations were performed in Wasilczyk Medical Clinic, Warsaw, Poland. All 25 patients who underwent positive clinical testing, ultrasound, and MRI examination that revealed ACL injury underwent knee arthroscopy with ACL reconstruction being conducted in the operating room. The procedures were documented using photographs and video recordings. 

Analytical material obtained from patients in the control group was documented in the form of MRI and ultrasound scans.

Four ultrasound parameters were evaluated:The inclination of the ACL.Swelling/scarifications of the ACL proximal attachment to lateral femoral condyle.Swelling/scarifications of the ACL/posterior crucial ligament (PCL) compartment with change of the morphology of the posterior joint capsule complex.Dynamic instability in anterior drawer test with range from 0 to 2 mm, 3 to 4 mm, and ≥5 mm.The following coding system was established for ACL pathologies assessment in ultrasound examination: 0/1 for parameters 1,2, and 3 and 0/1/2 for parameter no 4.

### 2.3. Statistical Analysis

The frequency of individual ultrasound findings was compared between patients with and without ACL injury using the Fisher exact test for 2 × 2 contingency tables. The association of age with the presence of individual symptoms and with ACL damage was evaluated using the Mann–Whitney U test and the Kruskal–Wallis ANOVA test. The statistical analysis was performed using the Statistica 13.3 software (TIBCO Software Inc., Palo Alto, CA, USA).

## 3. Results 

### 3.1. Part I—Anatomical Study 

Because of the lack of an edema after an ACL injury on the cadaveric specimens, only two parameters out of four were examined in this part of the study—instability and inclination angle.

In all cases, in dynamic ultrasound examination after an ACL injury we reported instability. In all cases except one, we reported an abnormal inclination angle of the ACL. The sensitivity for instability and inclination angle in that part of the study was 100% and 90 %, respectively. Sample pictures of an anatomical study are presented in Figure 1 and Figure 2. 

### 3.2. Part II Clinical Study 

The study included 24 men and 26 women. Among those with an injured ACL, men accounted for 50%, while among those without an ACL injury, men accounted for 46%. The difference in gender distribution was not statistically significant (*p* = 0.99).

Our study revealed no significant association between patient gender and the frequency of tested parameters. The prevalence of abnormal ACL inclination was 42% in women and 29% in men (*p* = 0.50). Similarly, swelling of the ACL intact to the femur condyle was found in 38% of women and 42% of men (*p* = 0.95). In the ACL/PCL compartment, swelling was observed in 38% of women and 46% of men (*p* = 0.81). Regarding dynamic instability of the ACL, it occurred in 54% of women and 67% of men, with the measurements exceeding 5 mm in 50% of women and 56% of men (*p* = 0.82).

Detailed results of the ultrasound assessments of the four parameters are presented in Table 1.

Abnormal inclination of the ACL in ultrasound was noted in 8% of patients without an ACL injury and in 67% of patients with an ACL injury (Figure 3 and Figure 4). Abnormal inclination of the ACL in ultrasound is more common in patients with an ACL injury (*p* < 0.0001). The odds ratio (OR) was 24 (95% CI and 4.50—127.96). 

Swelling of the ACL intact to femur condyle was more common in patients with an ACL injury than in those without an ACL injury (83% vs. 0%, respectively, and *p* < 0.0001) (Figure 5). OR = incalculable.

Swelling of the ACL/PCL on ultrasound was more common in patients with an ACL injury than in those without an ACL injury (88% vs. 0%, respectively, and *p* < 0.0001) (Figure 6 and Figure 7). OR = incalculable.

Dynamic instability was notably more prevalent in patients with ACL injuries (*p* < 0.0001). Among patients without an ACL injury, 23% exhibited dynamic instability within a range of 3–4 mm. In stark contrast, all patients with ACL injuries showed signs of dynamic instability: 33% of these patients had instability within the 3–4 mm range, while a 67% experienced instability ≥ 5 mm. 

The ultrasound test's ability to detect ACL damage based on abnormal ACL inclination has a sensitivity of 67% and a specificity of 92%. When assessing ACL damage through the presence of swelling in the ACL compartment using ultrasound, the test shows 83% sensitivity and 100% specificity. For detecting ACL damage by identifying swelling in the ACL/PCL compartment with ultrasound, the sensitivity is 88% and the specificity reaches 100%. The test’s effectiveness in diagnosing ACL damage through dynamic instability evaluation under ultrasound is 100% sensitive and 77% specific. It is important to note that in cases where patients with 3–4 mm and ≥5 mm abnormalities were categorized as injured, the sensitivity drops to 67% while specificity increases to 100% if only those in the ≥5 mm category are considered to be affected.

### 3.3. Decision Tree

Based on the statistical analysis, a decision tree for the diagnosis of an ACL injury from an ultrasound image was proposed (Figure 8). The decision tree created using the Statistica 13.3 software includes two modeling variables for the assessment of the morphology of the posterior joint capsule complex and dynamic instability (The choice of these two parameters is guided by the decision tree algorithm. The algorithm decides which variables best divide the study group, thereby yielding the best matching). Using the available database, the decision tree made only one incorrect matching, which translates to an accuracy of 98%. Based on the decision tree, it can be assumed that patients with the CL modeling of the posterior ligament complex have an ACL injury. In the remaining patients, dynamic instability should be additionally assessed. If it is ≥5 mm, then it can be assumed that an ACL injury is present.

## 4. Discussion

ACL is a most challenging ligament to visualize in an ultrasound examination of the human knee. This difficulty is due to its anatomical location—it is situated at the center of the knee joint. From the anterior ultrasound view, it is separated by subcutaneous tissue, the patellar ligament, and a thick layer of Hoffa’s body. Additionally, the co-occurrence of infrapatellar plica often makes the visualization of the ACL more complicated. From the posterior ultrasound view, the thick layer of popliteal fossa, posterior crucial ligament, and neurovascular bundle create even more difficulties in the ultrasound examination of the ACL view than they do from the anterior view. An MRI examination is a well-documented and widely accepted tool in orthopedic practice. However, ultrasound examination is also well documented and accepted in ACL examination. In my view, in orthopedic practice, ultrasound examination within detailed and carefully evaluated parameters offers invaluable opportunities to obtain additional information about the state of the ACL, which is crucial for making further clinical decisions. 

Sievert et al. [66] demonstrated that ultrasound is highly effective in ACL examination or even anteromedial (AM) bundle diameter measure and in his study he demonstrated that two researchers with low-to-moderate training with ultrasound measures can locate and measure the AM bundle of the ACL. The standard errors between sessions for Rater 1’s AM bundle and the ACL diameters were less than 0.03 cm. Intra-rater reliability was higher for the AM bundles compared to the full ACL, and was 0.76 vs. 0.59, respectively.

Chen et al. [67] have well documented the fact that ultrasound is a reliable tool for measuring ACL diameter compared to MRI examination. Additionally, they highlight three pathological findings for detecting ALC tears using indirect methods: a hypoechoic lesion at the femoral insertion of the ACL (sensitivity, 0.91 and specificity, 0.8), protrusion of the posterior fibrous capsule (sensitivity, 0.77 and specificity, 0.68), and S-shaped thickening of the posterior cruciate ligament (sensitivity, 0.65 and specificity, 0.71).

Also, Larsen [68] proved that ultrasound examination is a reliable tool for ACL diagnosis. The sonographic findings were confirmed in 59 out of 62 cases. The sensitivity was 88%, the specificity 98%, and the positive and negative predictive values 93 and 96%. Similar outcomes were demonstrated by other clinicians [69,70,71].

The presented study is the first to standardize and evaluate ACL state and propose a tree decision based on four independent ultrasound view parameters. Although other researchers evaluate similar parameters, the final diagnosis depends on their clinical expertise.

In the anatomical part of the study, we proved that an ACL injury destabilized the knee, with a dynamic ultrasound examination in all cases showing more than 5 mm instability in an anterior drawer test. Also, in all cases except one in this part of the study we found abnormal inclination of the ACL. For clear reasons, in this part of the study we exclude the other two parameters: swelling of the ACL/PCL posterior compartment and swelling of the ACL intact to femur condyle.

In the clinical part II of our study, we conducted a comparative analysis of four parameters between two groups: patients with a normal ACL and those with ACL injuries. Our findings revealed significant differences in these parameters between the two groups. Firstly, abnormal inclination of the ACL on ultrasound was observed in 8% of patients without ACL injury and in 67% of those with an injury; secondly, swelling of the ACL attachment to the femur condyle was exclusively noted in patients with ACL injuries (83% vs. 0%); thirdly, similar patterns were observed in swelling of the ACL/PCL compartment (88% vs. 0%); and finally, dynamic instability was significantly more prevalent in patients with ACL injuries, and this was noted in 23% of non-injured patients (all within a 3–4 mm range) and in all injured patients, with 33% showing 3–4 mm instability and 67% exceeding 5 mm. These results highlight the critical role of these ultrasound parameters in differentiating between normal and injured ACL states.

We developed a decision tree, which incorporates two key variables: the assessment of the morphology of the posterior joint capsule complex and dynamic instability. The selection of these particular parameters was driven by the decision tree algorithm, which determines the variables that most effectively differentiate the study group to achieve the best match. Our decision tree demonstrated a high level of accuracy, with only one incorrect match out of the total cases, resulting in an accuracy rate of 98%. According to this model, the presence of morphological changes in the posterior ligament complex is indicative of an ACL injury. For patients without such changes, dynamic instability should be further evaluated. If this instability exceeds 5 mm, an ACL injury can be presumed. This decision tree serves as a useful tool for diagnosing ACL injuries with high precision.

In the clinical part of the study, in two cases MRI outcome differs with arthroscopy diagnosis: the arthroscopy reveals an ACL tear. Detailed analysis of these two cases showed that in both cases dynamic ultrasound examination showed instability, and in one case abnormal inclination of the ACL view.

In the literature, there are many studies regarding ACL ultrasound examination including dynamic examination or evaluation of indirect methods like a hypoechoic lesion at the femoral insertion of the ACL, protrusion of the posterior fibrous capsule, or S-shaped thickening of the posterior cruciate ligament, but there are no articles reporting on the assessment of the ACL based on isolated ultrasound parameters.

Using the algorithm and available database, the tree decision in the clinical part of the study indicated an accuracy in the ultrasound examination of 98%.

A limitation of our study is that conducting an ultrasound examination, particularly when assessing the cross-section of the ACL at its femoral attachment site, demands considerable expertise and skill from the examiner.

## 5. Conclusions

The standardization of ultrasound examinations, based on four isolated parameters, may significantly enhance the sensitivity of imaging, and complements clinical and MRI diagnoses. These parameters include: (1) the inclination of the ACL; (2) swelling or scarring of the proximal attachment of the ACL to lateral femoral condyle; (3) swelling or scarring of the ACL/PCL compartment, accompanied by changes in the morphology of the posterior joint capsule complex; and (4) dynamic instability, classified into ranges of 0–2 mm, 3–4 mm, and ≥5 mm. This methodical approach not only aids in the precise assessment of the ACL’s condition but also helps in identifying subtle pathologies that might be overlooked in standard imaging techniques. By quantifying dynamic instability and evaluating morphological changes, such standardized ultrasound protocol could lead to earlier detection and more accurate treatment planning for ACL injuries.

## Figures and Tables

**Figure 1 diagnostics-14-00305-f001:**
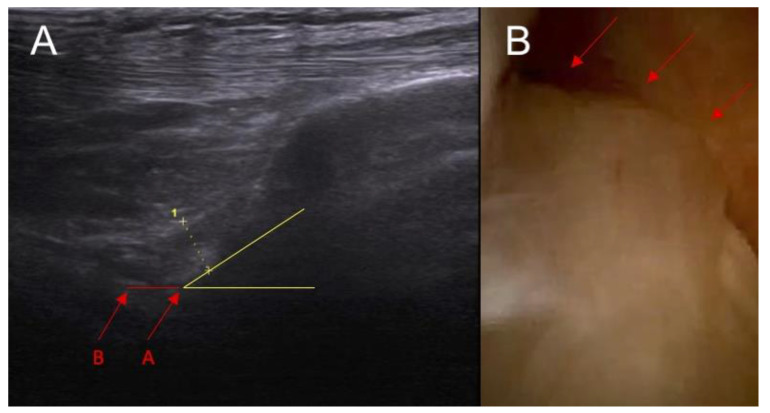
Anatomical part of the study. (**A**) Ultrasound examination: point A—baseline point of the ACL in the ultrasound image before the application of force and point B—the same point after the application of force in the anterior drawer test. Section A–B—actual functional instability on ultrasound examination. Yellow lines indicate the inclination of ACL. (**B**) arthroscopy view— arrows indicate intersection of the ACL.

**Figure 2 diagnostics-14-00305-f002:**
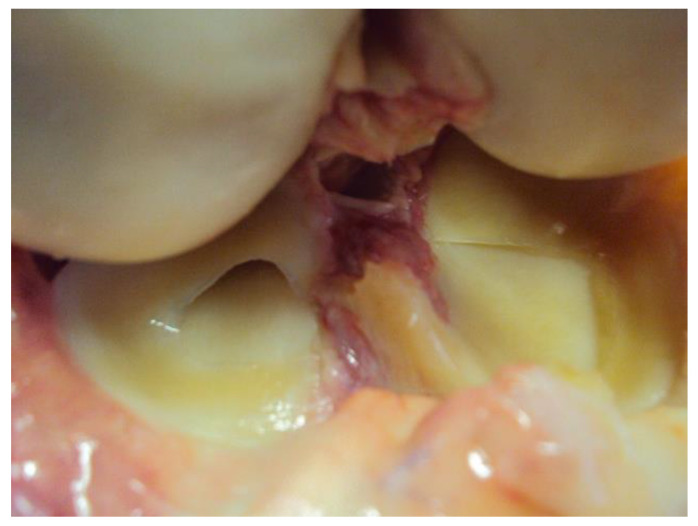
A preview image of intersection of the ACL in an anatomical specimen—arthrotomy.

**Figure 3 diagnostics-14-00305-f003:**
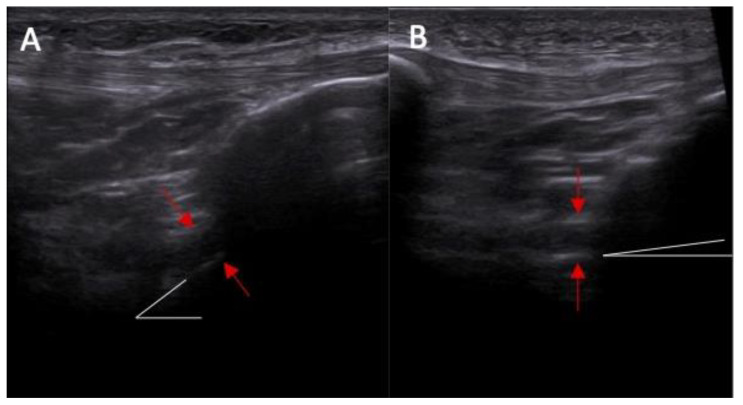
The inclination of the ACL—the ultrasound view (ACL is marked with red arrows). (**A**) normal = 45 degrees angle between extension of posterior border of the ACL and base line and (**B**) abnormal < 40 degrees angle between extension of posterior border of the ACL and base line.

**Figure 4 diagnostics-14-00305-f004:**
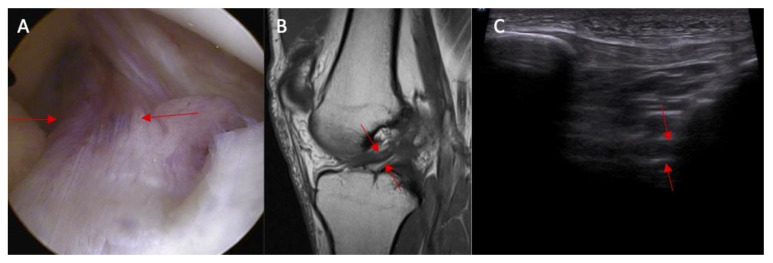
(**A**) ACL tear confirmed in arthroscopy view and MRI examination; (**B**) (marked with red arrows); and (**C**) abnormal inclination of the ACL in ultrasound view.

**Figure 5 diagnostics-14-00305-f005:**
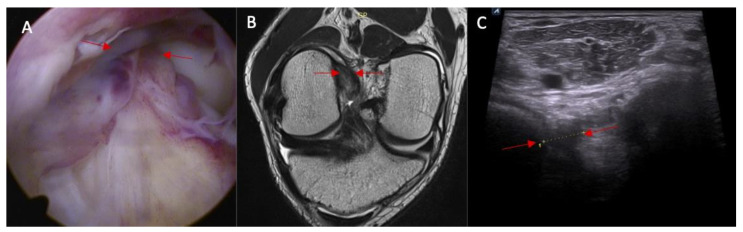
The proximal attached (marked with arrows) view of the ACL tear with edema. (**A**) arthroscopy view; (**B**) MRI view; and (**C**) ultrasound view—swelling of the ACL proximal attached to lateral femoral condyle.

**Figure 6 diagnostics-14-00305-f006:**
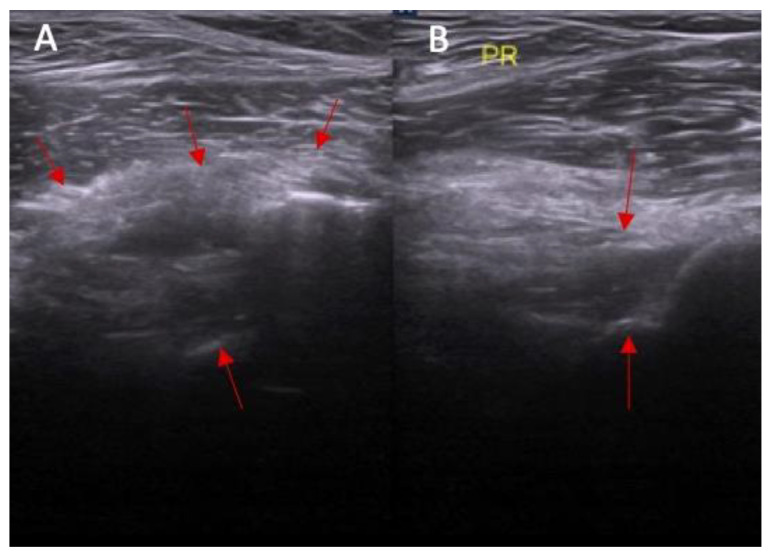
(**A**) Ultrasound view of swelling/scarifications of the ACL/PCL compartment with change of the morphology of the posterior joint capsule complex marked with the arrows and (**B**) normal view. PR—right.

**Figure 7 diagnostics-14-00305-f007:**
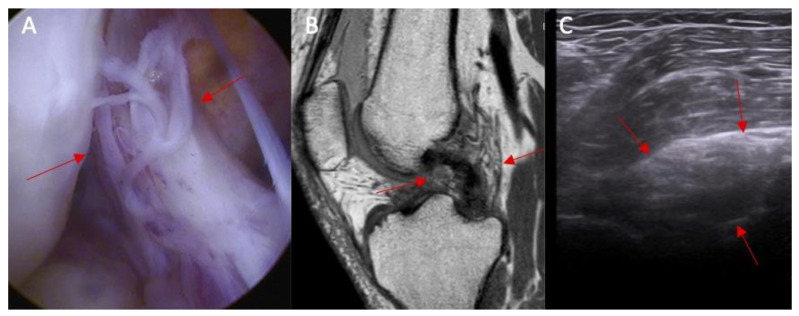
(**A**) ACL tear confirmed in arthroscopy view. Swelling of the ACL/PCL compartment with change of the morphology of the posterior joint capsule complex in (marked with red arrows) MRI view (**B**) and ultrasound view (**C**).

**Figure 8 diagnostics-14-00305-f008:**
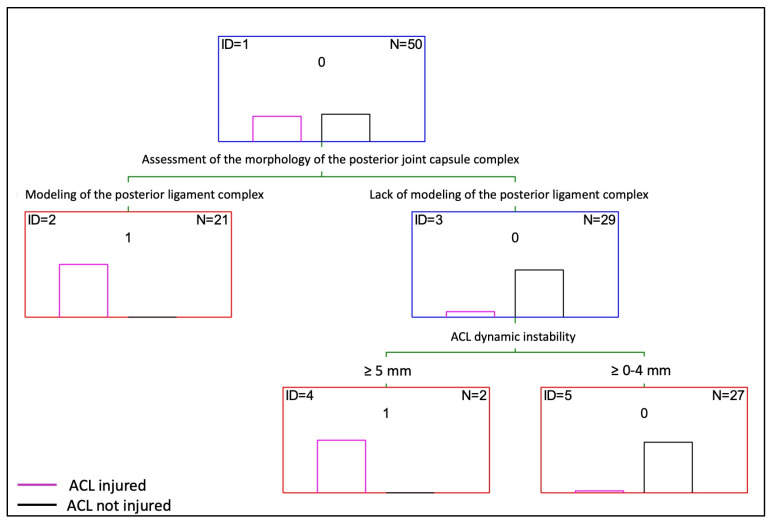
Decision tree of diagnosis of ACL injury in ultrasound image.

**Table 1 diagnostics-14-00305-t001:** Results of clinical parameters tested in ultrasound examinations.

Parameter	Study Group (*n* = 25)	Control Group (*n* = 25)	*p*
Abnormal inclination of the ACL (*n*, %)	17 (67%)	2 (8%)	<0.0001
Swelling of the ACL intact to femur condyle	21 (83%)	0	<0.0001
Swelling/scarifications of the ACL/PCL	22 (88%)	0	<0.0001
Dynamic instability:			<0.0001
0–2 mm	0	19 (76%)	
3–4 mm	8 (33%)	6 (23%)	
≥5 mm	17 (67%)	0	

ACL—anterior cruciate ligament and PCL—posterior cruciate ligament.

## Data Availability

The data underlying this article will be shared on reasonable request to the corresponding author.

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
