# Peer review of "The Value of Ultrasound Diagnostic Imaging of Anterior Crucial Ligament Tears Verified Using Experimental and Arthroscopic Investigations"

_diagnostics, 2024, doi:10.3390/diagnostics14030305_

Round 1
Reviewer 1 Report
Comments and Suggestions for Authors
Dear Authors,
The paper is very interesting for two reasons: 1st for its simplicity and 2nd for the type of lesion treated.
From the methodological point of view it is correct.
I would just like to make a few comments.
I assume that a sample size analysis has been performed. If so, I would like to know on what basis and how the patients were selected. Are there exclusion criteria?
In the introduction it is commented that it is a lesion more prevalent in women. It would be interesting to see if the decision tree (based on the statistical analyses) behaves equally well according to the sex of the patient.
It is possible that there are other variables that may confound the results such as age, type of activity, etc. Have they been taken into account?
No limitations paragraph. No limitations in this paper?
Author Response
Thank you for this review.
I assume that a sample size analysis has been performed. If so, I would like to know on what basis and how the patients were selected. Are there exclusion criteria?
Author: Yes, we calculated sample size and added description.
In the introduction it is commented that it is a lesion more prevalent in women. It would be interesting to see if the decision tree (based on the statistical analyses) behaves equally well according to the sex of the patient.
Author: We have added a statistical analysis of the influence of sex, and the difference in sex distribution was not statistically significant
It is possible that there are other variables that may confound the results such as age, type of activity, etc. Have they been taken into account?
Author: We based our analysis solely on the outcomes of ultrasound examinations. Therefore, these parameters will not confound the results
No limitations paragraph. No limitations in this paper?
Author: we have added limitations
Reviewer 2 Report
Comments and Suggestions for Authors
Nice study to address the ultrasound diagnostic value of ACL tears.
Please include figure to describe how to study instability and inclination angle in the anatomical study.
How is the sensitivity and specificity in the Part II clinical study?
Author Response
Thank you for this review.
In accordance with your suggestions, we have added new figures, and have addressed the specificity and sensitivity of Study Part II.
Round 2
Reviewer 1 Report
Comments and Suggestions for Authors
The authors have clarified all my doubts and comments. I have nothing else to say.